# Consumer Knowledge about Food Labeling and Fraud

**DOI:** 10.3390/foods10051095

**Published:** 2021-05-15

**Authors:** Maria João Moreira, Juan García-Díez, José M. M. M. de Almeida, Cristina Saraiva

**Affiliations:** 1Department of Veterinary Sciences, School of Agrarian and Veterinary Sciences, University of Trás-os-Montes e Alto Douro, 5001-801 Vila Real, Portugal; mjoaomoreira24@hotmail.com; 2Veterinary and Animal Research Centre (CECAV), University of Trás-os-Montes e Alto Douro, 5001-801 Vila Real, Portugal; juangarciadiez@gmail.com; 3Department of Physics, School of Sciences and Technology, University of Trás-os-Montes e Alto Douro, Quinta do Prados, 5001-801 Vila Real, Portugal; jmmma@utad.pt; 4CAP/INESC TEC, Technology and Science and FCUP, Faculty of Sciences, University of Porto, Rua do Campo Alegre, 687, 4169-007 Porto, Portugal

**Keywords:** consumer, food labeling, food confidence, consumer perception, food fraud

## Abstract

Food fraud is a growing problem and happens in many ways including mislabelling. Since lack of consumers’ knowledge about mandatory food labeling information and different types of food fraud may impact public health, the present work assesses consumers’ knowledge about these issues. Principal component analysis was performed to obtain a smaller number of uncorrelated factors regarding the usefulness and confidence of information displayed in food labels and the perception of food fraud. Results indicated that information displayed in food labels is useful, however the way it is presented may decrease consumer interest and understanding. Regarding respondents’ confidence in foodstuffs, over half of them stated that information provided in food labels is reliable. However, a lack of confidence about food composition is observed in those processed foodstuffs such as meat products. Food fraud is recognized by more than half of respondents with a higher perception of those practices that imply a risk to public health than those related to economic motivation. Age and education of consumers influenced the perception of the information displayed in the food labels, their confidence and knowledge about food fraud. Implementation of education programs to increase consumer knowledge about food labelling and fraud is essential. Respondents’ perception results could be use as guidelines by the food industry to improve food label design in order to enhance consumer understanding.

## 1. Introduction

Currently, new consumers’ issues related food safety have emerged in recent years related to aspects such as ingredients, allergens, chemical additives, food processes or health impact associated with long-term consumption among others [1]. Food labelling, compulsory by law in the European Union [2], has an important role in food safety. The indication of specific mentions such as ingredients, intended use, batch number, shelf-life and storage conditions are essential to guarantee food safety. Since food labeling influence consumers’ preferences at the time of purchase [3], consumers are advised to read the information displayed on labels to verify if the products meet their preferences and/or are adapted to specific nutrition programs (e.g., vegetarians) or adapted to specific health conditions (i.e., diabetics) [4]. Regarding food labeling, some studies assess consumers’ perception about nutritional labeling [5,6]. However, scarce research addresses the opinion and perception of consumers about the compulsory mentions displayed in food packages [7].

Some studies indicate that clear and legible information on food labels is the most appreciated characteristic by consumers [8]. However, little research has assessed consumers’ perception of that information [9]. Despite the effort made by the food industry to improve consumer information and adapt to the changes in food labeling defined by law [2], further research about the real perception of consumers regarding clarity and information understanding is still needed.

Food fraud is the act of purposely altering, misrepresenting, mislabeling, substituting or tampering with any food product at any point along the food chain. Since food fraud damages the food industry and decreases consumer confidence, some studies assessed the changes in the purchase behaviour [10,11]. Recent food fraud scandals have been associated with mislabeling practices. Although this practice is difficult to demonstrate, it seems there is an advantage in the low perception of consumers regarding food fraud and the scarce testing of foods for materials that are not expected to be present by food authorities. Thus, the objective of the present work is to evaluate consumers’ knowledge about food labeling and food fraud.

## 2. Materials and Methods

### 2.1. Survey Design and Data Collection

To assess consumer knowledge about food labeling and fraud, a specific online questionnaire was designed on Google forms and it comprised 42 questions (closed questions) divided into 5 groups based on the European food safety policy and scientific literature review. The questionnaire assessed: (i) opinion about the usefulness of information displayed in the whole food label of food products, (ii) respondent confidence about information displayed in food labels, (iii) respondent confidence regarding the constitution of food, (iv) respondent knowledge about consequences derived from food mislabelling, and (v) respondent knowledge about food fraud.

The survey distribution was mainly performed by email invitation and social media for a period of 12 months (September 2016–October 2017). E-mail addresses were collected from email lists of the University of Trás-os-Montes e Alto Douro (Portugal) including teachers and non-teacher personnel as well as email lists of students of the MSc in veterinary medicine and food engineering course. Appropriate information about the questionnaire structure and the objective of the research was provided to survey participants, allowing them to decide on their participation in this research study. Previously, 20 respondents were surveyed to detect potential errors in the questionnaire or difficulties on it interpretation among others that may bias the results. Questions concerning socio-demographic characteristics, such as sex, age, civil status, economic status, lifestyle and health of respondents were also included.

### 2.2. Data Analysis

Data from questionnaires were carefully checked and processed using SPSS 22.0 software (SPSS, IBM, Armonk, NY, USA). Data analysis was carried out as described elsewhere [12]. Briefly, Cronbach’s Alpha was calculated to estimate both the reliability and the consistency of the survey. The influence of the consumers’ socio demographic characteristics on understanding the food label information was assessed by a Kruskal–Wallis test and principal component analysis (PCA). The appropriateness of the PCA was checked by Bartlett’s sphericity test Kaiser–Meyer–Olkin criterion [13].

## 3. Results

### 3.1. Socio-Demographic Characteristics of Respondents

A total of 308 respondents answered the online survey. The sample set consisted of 83 men (26.9%) and 225 women (73.1%). 195 respondents (63.3%) were single, 21 (6.8%) divorced and 92 (29.9%) married. According to age, 23.5% were under 25, 62.0% ranged from 25 to 45 and 14.5% were over 45. Respondent salary was under 500 € (31.2%), 500–900 € (33.4%), 900–1500 € (21.1%) and over 1500 € (14.3%). Regarding respondent education and lifestyle, 81.2% were graduates, 95.8% declared a healthy lifestyle and 41.3% practised sport regularly. In addition, 116 (37.7%) respondents declared some dietary restriction while only 9 (2.9%) were vegetarians.

### 3.2. Assessment of the Opinion and Perceptions about Food Labelling and Food Fraud

Over 65% of respondents (Table 1) considered compulsory mentions in food labels useful although almost 65% of them declared some difficulty understanding the information displayed on them. In addition, some label characteristics such as font size, symbols or label design were considered negatively.

With respect to consumer confidence regarding the information displayed on food labels, only 52% of respondents stated that the information provided is reliable. However, over 60% of them declared that information indicated in food labels neither prevents food fraud nor guarantee the traceability. As regards of food constitution, over 55% of respondents declared distrust in the information provided by food manufacturers.

Among the different kinds of foodstuffs, respondents showed more confidence in less-processed products such as milk, oils or frozen products than in those subjected to more industrial processing such meat products, pre-cooked foods or ready-to-eat food products. However, consumers declared better confidence in foodstuffs labelled with protected designation of origin or protected geographical indication in the package. Regarding food fraud, over 75% of respondents indicated that mislabeling does not imply a health risk for consumers or an economic benefit for the food industry. Cronbach α values for all factors were larger than 0.87 and the internal consistency test based on the Cronbach α coefficient was 0.88, indicating good internal reliability.

### 3.3. Influence of Socio-Demographic Characteristics on Food Labelling Information

The influence of socio-demographic characteristics regarding food labeling and fraud perception are presented in Table 2. Age and education were the main factors that influenced (*p* < 0.05) the respondents’ perceptions about food labeling and fraud. Other socio-demographic characteristics such as sex, marital status, salary or life style were not significant (*p* > 0.05). Education influenced (*p* < 0.05) the usefulness of information displayed on food labels while age (*p* < 0.05) was related to the printed font size information on the label.

Respondents’ confidence in the information on food labels was influenced by age, mainly to health and nutritional aspects. Confidence about food fraud was influenced both by age and education while education influenced the knowledge about the food constitution. In contrast, no socio-demographic characteristics influenced the knowledge about the consequences of food mislabelling.

### 3.4. Respondents’ Knowledge about Food Fraud and Socio-Demographic Factors Influenced Knowledge

Results of respondents’ knowledge about food fraud related to perceived consequences for consumers are presented in Table 3. Overall, 55% of respondents showed some knowledge about food fraud. Among them, 61% recognized food fraud imply a risk for public health while only 51.8% associate this as an economic practice. Also, age and education influenced the knowledge of food fraud (*p* < 0.05). In addition, respondents who declared dietary restrictions or presented a healthy life style were more concerned with mislabeling of food ingredients.

### 3.5. Principal Component Analysis

Loadings of each principal component (PC) after the varimax and normalized rotation and communalities from the PCs are presented in Table 4. Figure 1 shows the projection of the 20 original variables on a two-dimensional space defined by the two PCs. The first and second PC together (PC1–PC2) accounted for 48.1% of the data variance. A total of variance was approximately 66.5%, achieved by using 5 PCs. The first component set the opinion regarding label information and perception of food mislabelling variables. The second component was characterized by the importance of food labelling application variables.

There is a significant association between the following variables: the information written on the labels is useful, Information displayed on front of the pack is useful and information displayed on the food label is easy to understand.

The variables which had the greatest partial contributions for variability were, in decreasing order, information displayed on front-of-the pack is useful (factor load (FL) = 0.77), food product is correctly described in the label information (FL = 0.77), overall information displayed on the food label is useful (FL = 0.74) and information displayed in food label is easy to understand (FL = 0.74), and for the positive dimension PC1. In contrast, mislabelling implies a risk to public health (FL = −0.33), mislabeling increase the consumer distrust (FL = −0.32) and mislabeling implies economic benefits for the food company (FL = −0.23) are located in a negative dimension.

PC2 found that food safety is associated with labelling and allows one to choose healthy foods, ensures nutritional quality and prevents fraud. In decreasing order, in a positive dimension of PC2, the variables information displayed in food label ensures food safety (FL = 0.80), information displayed in food label ensures nutritional quality (FL = 0.77), information displayed in food label prevents food fraud (FL = 0.74) and information displayed in food label is helpful for choosing healthy foods (FL = 0.71).

### 3.6. Spearman Test

Variables with some significant Spearman’s correlation coefficients (*p* < 0.01) are presented in Table 5. The risk to public health was positively correlated with mislabeling increases consumer distrust (r = 0.362, *p* < 0.001) and negatively correlated with overall information displayed in food label is useful (r = −0.291, *p* < 0.001), information displayed on the front of the pack is useful (r = −0.271, *p* < 0.001), food product is correctly described in the label information (r = −0.298, *p* < 0.001) and information displayed on the food label ensures nutritional quality (r = −0.208, *p* < 0.001). The loss of consumer confidence was mostly negatively correlated with information displayed on the back of the package is useful (r = −0.236, *p* < 0.001). In general, all opinions on label information variables were positively correlated in the confidence provided by food labelling variables.

## 4. Discussion

The new food policy framework established in the European Union is aimed not only to guarantee food safety but also to improve public health. Since consumer demands regarding healthy and safer products have increased in the last years, the importance of food labeling and the information provided for consumers has gained huge importance nowadays [14]. Also, the recent food fraud scandals, due to mislabeling practices, have increased consumer distrust in the food industry since information displayed on food labels does not reflect the real characteristics declared [15].

Food labeling can be considered as a communication tool between food industry and consumers. Although some reports highlighted the influence of food label information and purchase behavior [16], the lack of interest of consumers in food label have recently been discussed [17] and are mainly associated with a lack of time [12]. Our study showed that overall information is useful for respondents although specific mentions such as information located on the back of the package, symbols or proper description of food products seem to be not useful enough. Although the difficulty understanding of information displayed on food label was not affected by socio-demographic characteristics in the present study, age and education have been referred to in other works as critical factors in food label comprehension [18]. In addition, the reduced font size decreases consumer confidence both in the food company and food product quality

The importance of food label design, such as font colors and graphics, may affect consumer choice decisions [19]. The lack of usefulness declared by respondents could be associated with factors as text blocks at right angles, separation of the nutrition facts table from the list of ingredients, inadequate spacing between lines or words placed over illustrations [20].

Consumers’ confidence regarding food fraud has been affected in recent years due to the recent food scandals such as horse meat scandal in which horse meat was added to meat product instead beef [21]. This situation is compatible with the results observed in the present work in which meat and meat products inspired the worse confidence among respondents. In addition, it seems that lack of confidence is directly proportional to the degree of food processing. Thus, olive oil has been considered as one of the foodstuffs that generates more confidence among respondents. However, olive oil, together with honey, are one of the most falsified foodstuffs today [22,23].

With regard to quality labelling, foodstuffs under protected designation of origin or protected geographical indication schemes displayed better confidence among the respondents, in accordance with [24], probably associated with a greater perception of control by food authorities.

Food fraud is an old practice that has always existed but recently has become much more of an issue for consumers, thanks to viral news articles and social media [25]. Although slightly more than half of the respondents recognize some food fraud practices, implementation of education programs to increase consumer knowledge is essential since education, according to the results, seems to be a critical factor [26]. It is important to remark that food fraud is a parallel problem to food labeling since this cannot be prevented only by modifications of the mentions displayed in foods.

Despite the food fraud representing a problem outside the law, there are some characteristics of food fraud that can be minimized throughout food labeling. Thus, the introduction of the new policy in the European Union (EU) helps to prevent food fraud since it restrict the attribution of specific quality characteristics or nutritional and health claims. However, intentional food fraud, as the horse meat scandal, is just impossible to prevent just with modifications in the labeling policy.

Moreover, it has been suggested that highly educated consumers are more likely to distrust the information on food labels [11]. Thus, these consumers are more willing to use a device to validate food label content. In addition, it seems that respondents recognize better food fraud practices that imply a public health risk than those related to economically motivated adulterations. The recent food frauds carried out by the food industry have been related to mislabeling practices due to the use of lower value ingredients not declared on the label [27,28]. Although it is difficult to explain, these practices carried out by the food industry could have taken advantage of factors such as lack of knowledge of consumers, as previously indicated, together with the lack of control mechanisms of food authorities. However, detection of food fraud and mislabeling is still a difficult task as discussed above.

Research about consumers’ perception about food fraud are mainly focused on indications of the country of origin [29], traceability information [30] or implementation of new food safety strategies such as food defense plans throughout the food chain [31]. However, no research is available that evaluates consumers’ knowledge about food fraud. Thus, the respondents’ perceptions about food labelling and fraud observed in the current work could be used as guidelines by the food industry to improve food label design to enhance consumer understanding and usefulness. Also, addition of enough information in a clear way may enhance consumer trust in a in a period of loss of confidence in the food industry due to the recent food fraud scandals.

## 5. Conclusions

The recent food fraud scandals related to mislabeling practices have increased consumer distrust in the food industry since information displayed on food labels does not reflect the real characteristics declared.

The present study showed that compulsory information displayed in food labels is useful; however, the way the information is presented may decrease consumer interest and also make it difficult to understand information. Regarding consumers’ confidence about the information displayed on food labeling, over half of respondents stated that information provided is reliable. However, respondents showed a lack of confidence on food composition that decreases in processed foodstuffs such as meat products. Food fraud is recognized by over half of respondents as a practice that implies a risk to public health.

Age and education were the most important socio-demographic factors regarding food label perception, confidence in its information and also knowledge about food fraud. Thus, implementation of education programs to increase consumer knowledge about food labeling and fraud is essential. Since scarce research is available about consumer perceptions, food label information and food fraud, the respondents’ perceptions observed in the current work could be used as guidelines by the food industry to improve food label design in order to enhance consumer understanding and usefulness. Also, addition of enough information in a clear way may enhance consumer trust a in a period of loss of confidence in the food industry due to the recent food fraud scandals.

## Figures and Tables

**Figure 1 foods-10-01095-f001:**
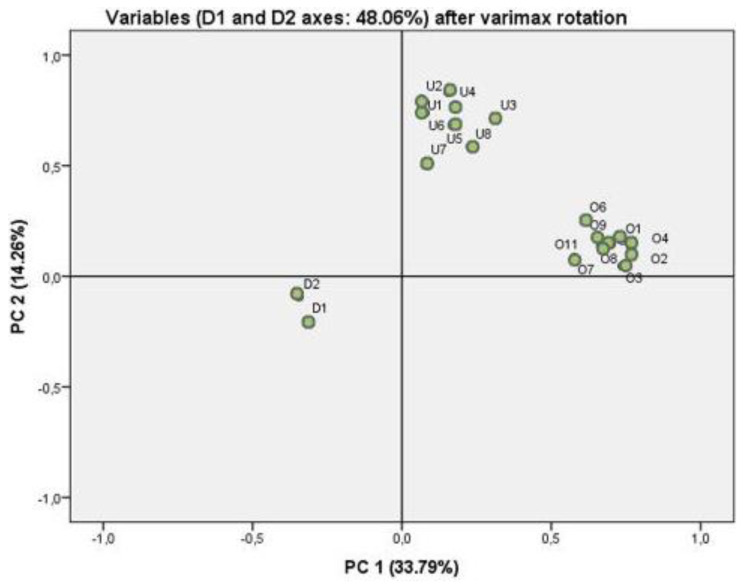
Loadings for the PC1–PC2 dimensions, after varimax normalized rotation, of the 20 variables selected to a principal components analysis: O1—The information written on the labels is useful; O2—It is a good source of information; O3—It is easy to understand the information on the label; O4—The label has information about the food product; O6—The label gives information about quality of the product; O7—The information on the back of the packaging is most useful; O8—Food labels contain enough information; O9—Symbols provide useful information; O11—The label is appropriate; U1—Ensures food quality; U2—Ensures food safety; U3—Lets you choose healthy foods; U4—Ensures nutritional quality; U5—Allows consumption according to ethics; U6—Prevents food fraud; U7—Respects religious beliefs; U8—Ensures traceability; D1—Risk to public health; D2—Loss of consumer confidence.

**Table 1 foods-10-01095-t001:** Consumers’ opinions and perceptions (%) about food labelling information and food fraud.

	Not Useful	Useful	Chro α
Respondent opinion about the usefulness of information displayed in food label			
Compulsory information displayed in food label is useful	34.3	65.6	0.88
Information displayed in food label is easy to understand	63.6	36.4	0.88
Food label design helps sell product	65.9	34.1	0.88
Information displayed in food label provide information about quality	61.7	38.3	0.88
Information displayed on front-of-the package is useful	45.7	54.3	0.88
Information displayed on back-of-the package is useful	53.0	47.0	0.88
Food label displayed enough information about the food	68.2	31.8	0.87
Food product is correctly described in the label information	59.8	40.5	0.88
Information about intended use and preparation mode is clear	48.7	51.3	0.88
Symbols displayed in food label provide useful information	58.8	41.2	0.88
Font size of is adequate	81.9	18.1	0.88
Food label design is appropriate	74.7	25.3	0.88
	**Not provides Confidence**	**Provides Confidence**	
Respondent confidence about information displayed in food label			
Information displayed in food label ensures food quality	43.9	56.1	0.87
Information displayed in food label ensures food safety	41.5	58.5	0.88
Information displayed in food label is helpful to choose healthy foods	26.9	73.1	0.87
Information displayed in food label ensures nutritional quality	37.3	62.7	0.88
Information displayed in food label allows consumption according to ethics	52.3	47.7	0.88
Information displayed in food label prevents food fraud	60.0	40.0	0.88
Information displayed in food label respects religious beliefs	70.5	29.5	0.88
Information displayed in food label ensures traceability	48.0	52.0	0.88
	**Not provides Confidence**	**Provides Confidence**	
Respondent confidence regarding the constitution of food			
Meat and meat products	71.7	28.3	0.88
Fish and fishery products	56.1	43.9	0.88
Milk and dairy products	40.6	59.4	0.88
Ready-to-eat products	75.3	24.7	0.88
Pre-cooked products	76.9	23.1	0.88
Frozen products	50.3	49.7	0.88
Olive oil and other oils	45.5	54.5	0.88
Foodstuffs with quality labels (PDO, PGI)	38.0	62.0	0.88
	**w/o Cons**	**Hard Cons**	
Respondent knowledge about consequences derived from food mislabelling			
Mislabeling implies a risk to public health	84.5	15.5	0.89
Mislabeling increase the consumer distrust	73.0	27.0	0.89
Mislabeling implies economic benefits for food company	75.0	25.0	0.89

w/o: without; cons: consequences; PDO: protected designation of origin; PGI: protected geographical indication; Chro α: Cronbach α.

**Table 2 foods-10-01095-t002:** Influence of socio-demographic characteristics in food label information, food label confidence and trust in food constitution (results *p* < 0.05 are statistically significant).

	Age	Education
Respondent opinion about the usefulness of information displayed in food label		
Overall information displayed in food label is useful	Ns	*p* < 0.05
Information displayed in food label is easy to understand	Ns	Ns
Information displayed in food label contains enough information about food product	Ns	*p* < 0.05
Food label design helps sell product	Ns	Ns
Information displayed in food label provide information about quality	Ns	Ns
Information displayed on front-of-the package is useful	Ns	Ns
Information displayed on back-of-the package is useful	Ns	Ns
Food label displayed enough information about the food	Ns	Ns
Food product is correctly described in the label information	Ns	Ns
Information about the intended use and preparation mode is clear	Ns	Ns
Symbols displayed in food label provide useful information	Ns	Ns
Font size of is adequate	*p* < 0.05	Ns
Food label design is appropriate	Ns	Ns
Consumer confidence about food label information		
Information displayed in food label ensures food quality	Ns	Ns
Information displayed in food label ensures food safety	Ns	Ns
Information displayed in food label is helpful for choosing healthy foods	*p* < 0.05	*p* < 0.05
Information displayed in food label ensures nutritional quality	*p* < 0.01	Ns
Information displayed in food label allows consumption according to ethics	Ns	Ns
Information displayed in food label prevents food fraud	*p* < 0.05	*p* < 0.05
Information displayed in food label respects religious beliefs	Ns	Ns
Information displayed in food label ensures traceability	*p* < 0.01	Ns
Consumer confidence regarding the constitution of food		
Meat and meat products	*p* < 0.01	Ns
Fish and fishery products	Ns	Ns
Milk and dairy products	Ns	*p* < 0.05
Ready-to-eat products	Ns	*p* < 0.05
Pre-cooked products	Ns	Ns
Frozen products	Ns	*p* < 0.05
Olive oil and other oils	Ns	Ns
Foodstuffs with quality labels (GDP, POD)	Ns	*p* < 0.01

Ns: not significant. PDO: protected designation of origin; PGI: protected geographical indication.

**Table 3 foods-10-01095-t003:** Respondents’ knowledge about food fraud and socio-demographic factors influenced knowledge.

	Knowledge		*p*			
	Known	Unknown	Age	Education	Dietary Restriction	Physical Activity
Risk to public health						
Presence of chemical hazards derived from food processes	60.4	39.6	ns	ns	*p* < 0.05	ns
Addition of unauthorized additives/preservatives	81.8	18.2	ns	ns	ns	*p* < 0.05
Addition of food additives/preservatives not declared on food label	65.6	34.4	ns	*p* < 0.05	ns	ns
Use of food approved additives/preservatives over the maximum levels defined by law	54.9	45.1	Ns	ns	ns	ns
Presence of genetically modified organisms not declared on food label	42.5	57.5	*p* < 0.05	ns	ns	ns
Economic gain						
Partial/total substitution of an ingredient/substance	65.9	34.1	*p* < 0.05	ns	*p* < 0.05	*p* < 0.05
Addition of unauthorized ingredient/substance	58.8	41.2	ns	*p* < 0.05	ns	ns
Use of authorized ingredient/substance over the maximum level defined by law	51.9	48.1	ns	*p* < 0.05	ns	ns
Frozen-thawed foods sold as fresh products	44.5	55.5	ns	ns	ns	ns
Adulteration of geographical origin of foodstuffs	52.9	47.1	ns	ns	ns	ns
Use of unauthorized food practices or processes	36.7	63.3	*p* < 0.05	*p* < 0.05	ns	ns

ns: not significant.

**Table 4 foods-10-01095-t004:** Factor loadings and communalities of variables in the first two components (PC1 and PC2) after varimax normalized rotation.

Variables	Factors Loading		
Bartlett’s Test of Sphericity	*p* < 0.001; df = 190; *X*^2^ = 2954.64		
KMO Measure	0.857		
	PC ^b^ 1	PC ^b^ 2	CM ^a^
Compulsory information displayed in food label is useful	0.74	0.16	0.76
Information displayed on front of the pack is useful	0.77	0.14	0.74
Information displayed in food label is easy to understand	0.74	0.05	0.62
Food product is correctly described in the label information	0.77	0.13	0.66
Information displayed in food label provide information about quality	0.61	0.26	0.53
Information displayed on back of the package is useful	0.59	0.09	0.41
Information about intended use and preparation mode is clear	0.70	0.16	0.73
Symbols displayed in food label provide useful information	0.65	0.16	0.56
Food label design helps sell product	0.67	0.14	0.60
Information displayed in food label ensures food quality	0.17	0.83	0.84
Information displayed in food label ensures food safety	0.09	0.80	0.79
Information displayed in food label is helpful to choosing healthy foods	0.32	0.71	0.70
Information displayed in food label ensures nutritional quality	0.18	0.77	0.66
Information displayed in food label allows consumption according to ethical values	0.19	0.69	0.60
Information displayed in food label prevents food fraud	0.10	0.74	0.60
Information displayed in food label respects religious beliefs	0.09	0.51	0.78
Information displayed in food label ensures traceability	0.25	0.59	0.60
Mislabeling implies a risk to public health	−0.33	−0.20	0.52
Mislabeling increase the consumer distrust	−0.32	−0.08	0.72
Mislabeling implies economic benefits for food company	−0.23	−0.14	0.75

CM—communality; PC—principal component; KMO—Kaiser–Meyer–Olkin. Df—differential factor.

**Table 5 foods-10-01095-t005:** Significant correlations of Spearman’s rho (*p* < 0.01) between variables.

Variables	O1	O2	O3	O4	O5	O6	O7	O8	O9	O10	O11	U1	U2	U3	U4	U5	U6	U7	U8	D1	D2
O1	1	0.775	0.556	0.573	0.209	0.366	0.380	0.389	0.386	0.275	0.375	0.318	0.299	0.390	0.258	0.237	0.164	0.062	0.293	−0.291	−0.210
O2	**0.000**	1	0.547	0.581	0.135	0.419	0.389	0.444	0.387	0.285	0.366	0.287	0.241	0.368	0.277	0.279	0.192	0.124	0.265	−0.271	−0.247
O3	**0.000**	**0.000**	1	0.569	0.135	0.312	0.312	0.407	0.354	0.460	0.459	0.151	0.162	0.327	0.215	0.210	0.157	0.151	0.254	−0.199	−0.229
O4	**0.000**	**0.000**	**0.000**	1	0.193	0.460	0.363	0.442	0.457	0.306	0.437	0.243	0.184	0.377	0.266	0.279	0.149	0.135	0.295	−0.298	−0.131
O5	**0.000**	0.017	0.018	**0.001**	1	0.187	0.390	0.187	0.228	0.193	0.192	0.119	0.034	0.058	0.093	0.173	0.100	0.097	0.172	−0.096	−0.194
O6	**0.000**	**0.000**	**0.000**	**0.000**	**0.001**	1	0.358	0.470	0.463	0.301	0.429	0.342	0.223	0.276	0.309	0.298	0.243	0.231	0.241	−0.208	−0.138
O7	**0.000**	**0.000**	**0.000**	**0.000**	**0.000**	**0.000**	1	0.443	0.291	0.267	0.342	0.187	0.142	0.281	0.191	0.263	0.133	0.113	0.145	−0.152	−0.236
O8	**0.000**	**0.000**	**0.000**	**0.000**	**0.001**	**0.000**	**0.000**	1	0.579	0.477	0.586	0.262	0.209	0.228	0.300	0.227	0.213	0.140	0.203	−0.142	−0.163
O9	**0.000**	**0.000**	**0.000**	**0.000**	**0.000**	**0.000**	**0.000**	0.000	1	0.407	0.449	0.233	0.171	0.260	0.242	0.222	0.214	0.171	0.290	−0.140	−0.123
O10	**0.000**	**0.000**	**0.000**	**0.000**	**0.001**	**0.000**	**0.000**	**0.000**	**0.000**	1	0.693	0.111	0.099	0.200	0.189	0.151	0.223	0.189	0.278	−0.090	−0.059
O11	**0.000**	**0.000**	**0.000**	**0.000**	**0.001**	**0.000**	**0.000**	**0.000**	**0.000**	**0.000**	1	0.229	0.156	0.244	0.284	0.253	0.201	0.157	0.252	−0.125	−0.079
U1	**0.000**	**0.000**	**0.008**	**0.000**	0.036	**0.000**	**0.001**	**0.000**	**0.000**	0.053	**0.000**	1	0.839	0.623	0.628	0.505	0.539	0.257	0.369	−0.232	−0.175
U2	**0.000**	**0.000**	**0.004**	**0.001**	0.551	**0.000**	0.012	**0.000**	**0.003**	0.083	**0.006**	**0.000**	1	0.572	0.559	0.449	0.533	0.192	0.346	−0.234	−0.095
U3	**0.000**	**0.000**	**0.000**	**0.000**	0.310	**0.000**	**0.000**	**0.000**	**0.000**	**0.000**	**0.000**	**0.000**	**0.000**	1	0.729	0.517	0.409	0.208	0.399	−0.270	−0.194
U4	**0.000**	**0.000**	**0.000**	**0.000**	0.103	**0.000**	**0.001**	**0.000**	**0.000**	**0.001**	**0.000**	**0.000**	**0.000**	**0.000**	1	0.530	0.461	0.257	0.434	−0.214	−0.133
U5	**0.000**	**0.000**	**0.000**	**0.000**	0.002	**0.000**	**0.000**	**0.000**	**0.000**	**0.008**	**0.000**	**0.000**	**0.000**	**0.000**	**0.000**	1	0.494	0.472	0.411	−0.144	−0.032
U6	**0.004**	**0.001**	**0.006**	**0.009**	0.081	**0.000**	0.020	**0.000**	**0.000**	**0.000**	**0.000**	**0.000**	**0.000**	**0.000**	**0.000**	**0.000**	1	0.421	0.464	−0.104	−0.107
U7	0.276	0.030	**0.008**	0.018	0.088	**0.000**	0.048	0.014	**0.003**	**0.001**	**0.006**	**0.000**	**0.001**	**0.000**	**0.000**	**0.000**	**0.000**	1	0.424	−0.047	−0.081
U8	**0.000**	**0.000**	**0.000**	**0.000**	0.002	**0.000**	0.011	**0.000**	**0.000**	**0.000**	**0.000**	**0.000**	**0.000**	**0.000**	**0.000**	**0.000**	**0.000**	**0.000**	1	−0.260	−0.079
D1	**0.000**	**0.000**	**0.000**	**0.000**	0.091	**0.000**	0.007	0.013	0.014	0.116	0.029	**0.000**	**0.000**	**0.000**	**0.000**	0.011	0.068	0.408	**0.000**	1	0.362
D2	**0.000**	**0.000**	**0.000**	0.021	**0.001**	0.015	**0.000**	**0.004**	0.032	0.301	0.168	0.002	0.097	**0.001**	0.019	0.581	0.060	0.157	0.169	**0.000**	1

O1—Compulsory information displayed in food label is useful; O2—Information displayed on front-of-the pack is useful; O3—Information displayed in food label is easy to understand; O4—Food product is correctly described in the label information; O5—Food label design helps sell product; O6—Information displayed in food label provide information about quality; O7—Information displayed on the back-of-the package is useful; O8—Information about intended use and preparation mode is clear; O9—Symbols displayed in food label provide useful information.; O10—Font size is adequate; O11—Food label design adequate; U1—Information displayed in food label ensures food quality; U2—Information displayed in food label ensures food safety; U3—Information displayed in food label is helpful to choose healthy foods; U4—Information displayed in food label ensures nutritional quality; U5—Information displayed in food label allows consumption according to ethics; U6—Information displayed in food label prevents food fraud; U7—Information displayed in food label respects religious beliefs; U8—Information displayed in food label ensures food traceability; D1—Mislabeling implies a risk to public health; D2—Mislabeling increase the consumer distrust; significant correlations (*p* < 0.01) and correspondent r values were presented with bold letter.

## Data Availability

The data that support the findings of this study are available from the corresponding author (Cristina Saraiva), upon reasonable request.

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
