# Peer review of "Consumer Knowledge about Food Labeling and Fraud"

_foods, 2021, doi:10.3390/foods10051095_

Round 1

Reviewer 1 Report

The aim of this paper is to assess the consumer’s knowledge about food labelling and food fraud. For this purpose, the authors made an on-line questionnaire, composed by 42 questions regarding, in brief, the usefulness and the responding confidence of consumers regarding food label, and the knowledge regarding food mislabelling and fraud. The authors analysed the data collected from the questionnaire, by the use of statistical instruments. In the results, the authors highlight the importance of a clear label and food presentation to lead to the product purchase, and what are the most important socio-demographic characteristics influencing the perception of food labelling. In addition, they demonstrated how a foodstuff with PDO or PGI trademark can generate more consumer’s confidence.  

Broad comments I think that writing the questions of the questionnaire in italics along the manuscript text could improve their comprehension by the readers.

Specific comments

P2 l 73 would it be possible to explain better on the basis of what the email addresses have been selected? 

P7 Figure 1 symbols uses in the PC analysis can be confused with the variables labelled with the letter O. I suggest to use another kind of symbol.

P10 L 221 the horse meat scandal should already be explained here, rather than later (L 242)

Author Response

The aim of this paper is to assess the consumer’s knowledge about food labelling and food fraud. For this purpose, the authors made an on-line questionnaire, composed by 42 questions regarding, in brief, the usefulness and the responding confidence of consumers regarding food label, and the knowledge regarding food mislabelling and fraud. The authors analysed the data collected from the questionnaire, by the use of statistical instruments. In the results, the authors highlight the importance of a clear label and food presentation to lead to the product purchase, and what are the most important socio-demographic characteristics influencing the perception of food labelling. In addition, they demonstrated how a foodstuff with PDO or PGI trademark can generate more consumer’s confidence.  

RE: Thank you very much for the revision which helped us to improve the quality of the manuscript. The entire manuscript was newly revised according the reviewer´s suggestions.

1) Broad comments: I think that writing the questions of the questionnaire in italics along the manuscript text could improve their comprehension by the readers.

RE: The questions of the questionnaire were presented in italics as suggested.

2) Specific comments

P2 l 73 would it be possible to explain better on the basis of what the email addresses have been selected? 

RE: e-mail addresses were collected from email lists of the University of Trás-os-Montes e Alto Douro including teachers and non-teacher personnel, as well as email lists of students of the MSc of Veterinary medicine and food engineering. This information was added to the text as suggested by the reviewer.

3) P7 Figure 1 symbols uses in the PC analysis can be confused with the variables labelled with the letter O. I suggest use another kind of symbol.

RE: we agree with the reviewer comment. To improve the understanding of the figure, dots were coloured.

4) P10 L 221 the horse meat scandal should already be explained here, rather than later (L 242)

RE: the explanation of horse meat scandal was indicated in line 221 as suggested by reviewer.

Reviewer 2 Report

The present research carries out a study of the knowledge of consumers about food labeling and how this affects their perception of them and their subsequent purchase, as well as their knowledge about food fraud.  I consider that the article provides the necessary information to understand the analytical problem posed for this research.

I consider the results obtained are of high interest to the scientific community and that can be employed for the different food industries or even for government institutions to regulate the labeling designs and changes.

I only have a couple of comments:

-I have detected some minor errors in English spelling, I highly recommend reviewing the English by the authors.

-The authors mentioned in the different moments the "socio-demographic characteristics" but I think that they are only selected consumers for one zone of the country or of differents zones?. Please, clarify it.

Author Response

Reviewer 2

The present research carries out a study of the knowledge of consumers about food labeling and how this affects their perception of them and their subsequent purchase, as well as their knowledge about food fraud.  I consider that the article provides the necessary information to understand the analytical problem posed for this research. I consider the results obtained are of high interest to the scientific community and that can be employed for the different food industries or even for government institutions to regulate the labeling designs and changes.

RE: The authors would like to thank the Reviewer for the valuable comments and the positive feedback of the present research.

I only have a couple of comments:

1) I have detected some minor errors in English spelling, I highly recommend reviewing the English by the authors.

RE: the entire manuscript was revised and corrected throughout.

2) The authors mentioned in the different moments the "socio-demographic characteristics" but I think that they are only selected consumers for one zone of the country or of different zones? Please, clarify it.

RE: As indicated in text, consumers questioned included university personnel, as well as students from MSs of veterinary medicine and food engineering. Most of student come from different locations of Portugal. Also, dissemination of questionnaire by social media contributes to increase the geographic variety of respondents.